# Microbial Responses to Various Types of Chemical Regents during On-Line Cleaning of UF Membranes

**DOI:** 10.3390/membranes12100920

**Published:** 2022-09-23

**Authors:** Zeyuan Gao, Qiuying Chen, Xiaolan Song, Jingwei Wang, Weiwei Cai

**Affiliations:** 1School of Chemistry and Chemical Engineering, Beijing Institute of Technology, Beijing 102488, China; 2School of Environment, Beijing Normal University, Beijing 100875, China

**Keywords:** on-line membrane cleaning, microorganisms, ultrafiltration, membrane biofouling, biofilm

## Abstract

Ultrafiltration is widely used to treat various environmental waters, and on-line membrane cleaning with various chemical reagents is frequently employed to sustain the filtration flux. However, the residue of cleaning agents in the ultrafiltration system is unavoidable, which may affect microbiological properties and biofilm formation during the next-round filtration. By investigating the changes in microbial characteristics, and their biofouling behaviors after exposure to HCl, NaOH, NaClO, citric acid (CA), and sodium dodecyl sulfonate (SDS), this study fills a knowledge gap in microbial responses to various types of chemical cleaning agents in an ultrafiltration system. The result shows that HCl, NaOH, and NaClO affect the bacterial properties and subsequent attachment on the membrane surface, while CA and SDS have no obvious influence on microorganisms. Specifically, HCl, NaOH, and NaClO reduce the hydrophobicity and mean size of suspended microorganisms, increase the extracellular polymeric substances (EPS) release, and trigger intracellular reactive oxygen species (ROS) generation, resulting in the death of a large quantity of microorganisms. Due to the self-protecting strategy, plenty of living cells aggregate on the membrane surface and form a cake layer with a stratified structure, causing more severe membrane biofouling.

## 1. Introduction

Membrane separation technology has been well-developed since it emerged in the 20th century, and widely applied in water treatment, dairy product production, and medical industries. Ultrafiltration, as one of the low-pressure membrane filtration processes, received a lot of attention [1,2]. However, membrane fouling is still a primary obstacle restricting large-scale application. Foulants deposited on membrane surfaces or into the pores could cause the increase in trans-membrane pressure (TMP) and the decrease in the permeate flux of the UF process, resulting in higher operating energy consumption and cleaning frequency [3].

Based on the biological and chemical characteristics, membrane fouling can be classified into inorganic fouling, organic fouling, and biofouling [4], among which biological fouling is a long-standing problem. The forming of biofouling is a complex, dynamic, and relatively slow process, which can be derived from the attachment of microorganisms or microbial products on the membrane surfaces [4]. Although some useful methods are proposed, such as adjusting the operating conditions of the ultrafiltration system [5,6], selecting antibacterial membranes [7], etc., to reduce biofouling, there is still a lack of a long-term, valid anti-fouling technique [8]. A long-term running of a UF system still needs membrane cleaning to restore its flux during the operation [4].

The selection of membrane cleaning agents is based on different fouling conditions. Usually, on-line chemical cleaning uses a variety of chemical cleaning agents, in order to achieve the purpose simply and efficiently [4,9,10]. Specifically, acid cleaning mainly removes inorganic fouling by relying on neutralization and metathesis reactions [4,11]. Alkaline cleaning can decompose large particles into smaller particles or soluble substances [12]. Oxidants primarily oxidize the functional groups of foulants to carboxyl groups, ketone groups, or aldehyde groups with hydrophilic properties, which are more easily hydrolyzed and removed at a suitable pH value. Chelating cleaning agents form a complex with foulants through physical and chemical reactions, and citric acid (CA) could affect biofilm formation by removing specific minerals and enzymes [13]. Surfactants with both hydrophilic and hydrophobic groups can remove contaminants by reducing their adhesion on the membrane surface [14].

During the on-line cleaning process, the membrane module remains in the reactor. The cleaning agent is injected into the membrane module through the permeation side, and chemically reacts with foulants on the membrane after passing through the membrane pores [15,16]. Afterwards, the residual effective chemical cleaning agent further diffuses into the entire reactor, thereby affecting the properties of microorganisms in the system. When the on-line membrane cleaning is finished, the UF system restarts, accompanied by these affected microorganisms attaching onto the membrane again [17]. This means membrane chemical agents not only remove the fouling, but also trigger side effects in the UF system [18,19]. In the literature, many studies pay attention to the effect of NaClO on activated sludge in membrane bioreactors (MBR), finding that NaClO could induce great changes in the properties of activated sludge such as cell dissolution, extracellular polymeric substances (EPS) release, intracellular reactive oxygen species (ROS) generation, etc. [20,21]. A previous study shows that NaClO promotes the release of dissolved organic carbon from activated sludge, while NaOH causes the release of soluble nitrogen [18]. However, there is still no related study focusing on the UF system, where abundant aqueous microorganisms exist.

According to the actual on-line cleaning processes, oxidants, acid–base reagents, surfactants, and chelating agents are commonly used for UF membranes [4,11], potentially resulting in different influences on aqueous microbes. In this study, five different types of chemical cleaning agents, i.e., NaClO, NaOH, HCl, CA, and sodium dodecyl sulfonate (SDS), were employed in order to comparatively explore the microbial responses to them and their influences on biofouling formation. The achieved results should provide some valuable information for optimizing current chemical cleaning strategies.

## 2. Materials and Methods

### 2.1. Cultivation of Microorganisms

Coliform bacteria are often used as the indicator organisms for microbial contamination in water bodies [17,22,23], and in this study, *Escherichia coli* was also selected as the representative of aqueous microorganisms present in the UF system [23]. The test *Escherichia coli O157:H7* (CGMCC 1.12873, Beijing, China) was grown in Luria–Bertani (LB) broth at 37 °C overnight. After being centrifuged (5500 rpm, 10 min) and washed with 10 mM phosphate buffer saline (PBS) solution three times, the microorganisms were resuspended in PBS solution and diluted to a final concentration of 10^5^ CFU/mL, which was in the range of typical microbial concentration in surface water [24,25].

### 2.2. Chemical Treatment Assay

Since the purpose of this study was to explore and compare the microbial responses to various types of membrane cleaning, a typical and commonly used concentration was selected for each of the five chemical reagents according to the literature [10,12,15,26,27]. Specifically, 0.2 mol/L HCl or NaOH was used to adjust the pH of the *E. coli* suspension to 2.0 and 12.0, respectively; for CA, SDS, and NaClO, each reagent was directly added to the cell suspension separately until the final concentration reached 300 mg/L CA [10,26], 300 mg/L SDS [12,27], and 100 mg/L NaClO [15]. All these chemicals were purchased from Macklin Company. Before usage, NaClO was standardized by N, N-diethyl-p-phenylenediamine (DPD) chorine reagent (Hach, Loveland, CO, USA). The whole reaction was continuously stirred at 300 rpm for 1 h to ensure sufficient contact between cleaning agents and microorganisms. A blank experiment without chemical addition was also performed as a control. To eliminate the influence of residual chemical reagents on subsequent experiments, the microorganisms after treatment were collected on a 0.45 μm filter (Jinteng, Tianjin, China) by low suction pressure of 0.03 MPa, then resuspended in fresh PBS solution, and the concentration was adjusted to ensure consistent biomass in the following cross-flow filtration tests [28,29,30].

### 2.3. Cross-Flow Ultrafiltration Test

The fouling propensities of microorganisms after reacting with different cleaning agents were evaluated in a standard cross-flow ultrafiltration system operated at a constant pressure of 0.15 Mpa, as shown in Figure 1. The membrane cell (CF042, Sterlitech, Auburn, WA, USA) had outer dimensions of 12.7 × 10 × 8.3 cm and a feed channel depth of 0.23 cm. A flat-sheet hydrophilic polyethersulfone (PES) membrane (Beijing Saipuruite, China) with a nominal molecular weight cut off (MWCO) of 50 kDa, mean pore size of 12.18 nm, and zeta potential of −3.06 mV (pH = 7.0) was used in the system. The zeta potential of membrane was measured at pH 7.0 by SurPASS (Anton Paar, Graz, Austria), which was consistent with our previous work [3,15,31]. Prior to use, the PES membrane was soaked in deionized water for 24 h. After pre-compaction by filtrating deionized water at 0.15 MPa and 25 ± 1 °C for 1 h, 1 L microbial suspension was introduced into the system by a pre-calibrated digital gear pump (Cole-Parmer, Chicago, IL, USA) at the recirculation superficial liquid velocity of 0.1 m/s. During filtration, the permeate was recycled back into feed water to avoid the significant enrichment of bacteria. Labview software was used to record the value of the balance (Cole-Parmer, Chicago, IL, USA) every 3 min, and automatically calculate the flux *J* of the system. The operation was terminated after 180 min.

### 2.4. Determination of EPS and Intracellular ROS of Microorganisms

The extraction and determination of polysaccharides (PS) and proteins (PN) in EPS were carried out according to a previous study [19]. To extract EPS from microorganisms, 20 mL of suspended microbial samples after reaction with cleaning agents was centrifuged at 10,000 rpm for 10 min, and then resuspended in an equal volume of PBS solution. The mixture was kept in an 80 °C water bath for 30 min, followed by a 1 min vortex. After centrifugation, samples for testing PS and PN were finally obtained by filtering the supernatant through a 0.45 μm membrane filter. PS was tested by the phenol–sulfuric acid method [32], and PN was tested by the Folin phenol method [33]. Glucose and bovine serum albumin (Macklin, Shanghai, China) were selected as standard samples. Each test was repeated more than 3 times.

The intracellular ROS content was tested according to the protocol provided by the reactive oxygen species fluorescent probe kit (Beyotime, Shanghai, China). Specifically, the samples were supplemented with 2,7-dichlorodihydrofluorescein diacetate (H2DCFDA) at a working concentration of 10 μM and incubated at 37 °C for 30 min in dark, then washed three times with PBS to remove unabsorbed fluorescence probe. Subsequently, 200 µL of the prepared samples were transferred to a 96-well flat bottom plate, and the fluorescence values were tested using a multifunctional microplate reader (BioTek, Winooski, VT, USA). The excitation wavelength was set at 488 nm and the emission wavelength was set at 525 nm. Each test was repeated more than 6 times.

### 2.5. Determination of Microbial Viability and Visualization of Biofilm

The viabilities of suspended and membrane-attached microorganisms were determined according to the protocol provided by the Baclight Viability Assay Kits (Thermo Fisher, Waltham, MA, USA). Briefly, the microorganisms attached to the membrane were gently resuspended in PBS solution, with their biomass content adjusted to remain consistent with the suspended samples. Afterwards, each 100 µL sample was mixed with 100 µL of diluted SYTO 9 and PI mixed dyes in a 96-well plate, then placed at room temperature for 15 min in the dark. The fluorescence intensities of the excitation/emission wavelengths of 485/530 nm (green, reflects viable bacteria) and 485/630 nm (red, reflects dead bacteria) were tested through a microplate reader (BioTek, Winooski, VT, USA). The relative viability of microorganisms was reflected by the ratio of green to red fluorescence intensity. A series of standard microbial suspensions with different viabilities were prepared by mixing live and dead bacteria. Each sample was tested at least six times.

In addition, the viability of microorganisms on the membrane surface was further visualized by a confocal laser scanning microscopy (FV1000, Olympus, Tokyo, Japan). To prepare samples, 100 µL of SYTO 9 and PI mixed dyes was dropped on the surface of biofilm with a rough area of 1 × 1 cm^2^. Then, the samples were left to stand in the dark for 15 min. After the excess dyes were sufficiently washed by PBS, the biofilm structure was fully observed by CLSM.

### 2.6. Determination of Microbial Surface Properties

According to the method proposed by Busscher et al. [34], the contact angles of microbial cells were measured to reflect the changes in surface hydrophobicity. Specifically, the microbial suspension was filtrated to form a distinct cell layer, and then dried in a desiccator to obtain samples. A goniometer (OCA 20, Dataphysics, Filderstadt, Germany) was used to test the water contact angle of microbial samples, and each measurement was performed at least 8 times. In addition, the zeta potential and mean size of microorganisms in suspension were tested by a Nanosizer (Malvern, Worcestershire, England), and each test was repeated more than 3 times to obtain the average value.

## 3. Results and Discussion

### 3.1. Properties of Microorganisms after Exposure to Different Chemical Cleaning Agents

The results in Figure 2 show that none of the five cleaning agents significantly affect the zeta potential of the microorganisms. The observed variation is not obvious, and may not distinctly affect the subsequent biofouling formation by the microorganisms. However, the contact angle and mean particle size change differently. HCl, NaOH, and NaClO significantly decrease the water contact angle of the microbial surface, suggesting an increase in the hydrophilicity of the cell surface. On the other hand, CA and SDS have minimal impact on surface hydrophilicity. As for the particle size of microorganisms, it seems that only CA has no obvious effect. The other four cleaning agents greatly reduce the mean size of microorganisms. Among them, the impact of SDS is the most obvious, and the influence of HCl and NaOH is relatively weak [35,36].

### 3.2. EPS and ROS Productions after Exposure to Different Chemical Cleaning Agents

EPS, which is mainly composed of polysaccharides (PS) and proteins (PN), has an indelible effect on microbial aggregation and plays a key role in biofilm formation [19]. The microorganisms affected by CA and SDS release roughly the same EPS content as the microorganisms without treatment, reflecting the limited effect of these two cleaning agents on EPS production. However, the concentrations of PS and PN upon exposure to HCl and NaOH increase significantly, leading to the total EPS increasing by 232.9% and 241.6%, respectively, as compared with the blank. After being exposed to NaClO, it is noteworthy to find that the concentration of PS decreases, while the concentration of PN increases remarkably, resulting in a total increase of 364.1% in the EPS content. However, it is reported that, compared with PN, PS may contribute to denser biofilm formation, leading to more severe membrane fouling [37,38]. Figure 3b further shows the amount of the intracellular ROS content of microorganisms after exposure to different chemical cleaning agents, which are produced in large quantities because of the attacks of various chemicals, probably damaging cellular DNA, RNA, and proteins [16,39]. It could be clearly found that, compared with the blank, HCl, NaOH, and NaClO trigger a significant overproduction of oxidative stress in the surviving bacteria. According to existing studies [16,19], the formation rate of biofilm is positively correlated with intracellular ROS levels, thereby the three chemical cleaning agents may aggravate subsequent biofouling formation during filtration. Comparatively, CA and SDS have minimal and negligible effects on ROS production.

### 3.3. Membrane-Fouling Potentials of Microorganisms after Exposure to Different Chemical Cleaning Agents

Figure 4 reflects the decrease in the flux of the UF membrane caused by the microorganisms after reaction with different chemical cleaning reagents. Obviously, the flux curves of the microorganisms affected by CA and SDS almost coincide with that of the original microorganisms. The severity of membrane fouling caused by the other three cleaning agents is in the order of NaOH > HCl > NaClO. As compared with the smooth curve of the original microorganisms, the microorganisms exposed to HCl, NaOH, and NaClO cause a fast and significant flux decrease in the early stage of filtration. It is believed that it is difficult for the original microorganisms, without external interruption, to form a significant cake layer on the membrane surface because of the high shear force from the cross-flow system [19,40]. However, HCl, NaOH, and NaClO may stimulate the surviving microbes to aggregate and form biofilm, which is recognized as a self-protecting strategy under harsh conditions [16]. The membrane surface images after filtration in Figure 5 also display consistent results. The membrane surfaces of microorganisms affected by CA and SDS seem similar to those of original microbes and the virgin membrane, indicating less severe microbial adhesion during 180 min filtration. For microorganisms affected by HCl, NaOH, and NaClO, thick fouling layers are clearly observed, and the surfaces of HCl and NaOH are relatively uneven compared to the smooth layer formed under the effects of NaClO. This might be explained by the capability of NaClO to decompose microbial aggregates to smaller sizes, as evidenced by Figure 2a.

### 3.4. Morphology of Biofilm Layer

After filtration, the morphology and structure of the biofilm on the membrane surface were studied. As shown in Figure 6, after exposure to different types of chemical cleaning reagents, the ratios of live and dead cells in suspension vary. Specifically, CA and SDS seldomly affect the survival of microorganisms, whose survival rate remains above 94.2%. However, the proportion of surviving microorganisms decreases to 30.5%, 18.7%, and 12.4% upon exposure to NaClO, HCl, and NaOH, respectively. After filtration, the ratios of live cells attached to the UF membrane dramatically increase to 85.4%, 74.2%, and 70.2% for NaClO, HCl, and NaOH, respectively, while the living cell proportion remains similar to that of the suspended microorganisms for CA and SDS. These results again demonstrate that more surviving microbes tend to attach onto the membrane under unfavorable conditions such as NaClO, HCl, and NaOH. CA and SDS exert little influence on microorganisms, thus, no biofouling aggravation is observed.

The CLSM images in Figure 7 display the biofilm structure formed by the microorganisms under various situations. The biofilm morphologies affected by CA and SDS are basically the same, which are all stacked structures composed of living cells. However, under the influences of NaOH, HCl, and NaClO, the structures are stratified with plenty of living cells aggregated on the bottom layer and dead cells appearing on the top. Comparatively, the upper layer of the biofilm affected by NaOH contains more dead cells than HCl and NaClO.

Based on the above results, exposure to NaOH, HCl, and NaClO increases EPS release (Figure 3a) and triggers the overproduction of intracellular ROS (Figure 3b), which is related to the self-protecting strategy of microorganisms [16]. During filtration, the survived microorganisms tend to rapidly colonize onto the membrane surface, assisted by abundantly released EPS, eventually forming a living cell layer covered by dead cells to resist the further threat of chemical agents (Figure 7), which is similar to the observations by Cai et al. [16,19]. Notably, the severity of membrane biofouling caused by various cleaning agents seems closely corelated to the intracellular ROS production and PS content in the released EPS. For example, under the threat of NaOH, microorganisms with the highest intracellular ROS production and PS release cause the most serious membrane fouling. In contrast, microorganisms exposed to NaClO release more PN than PS, as well as demonstrating lower ROS production, finally presenting a mitigated biofouling development (Figure 4).

## 4. Conclusions

The current work studied the effects of different membrane cleaning agents on bacterial characteristics and subsequent biofilm formation during ultrafiltration. The main findings can be summarized as follows:(i)All of five cleaning agents do not significantly affect the zeta potential of microorganisms. HCl, NaOH, and NaClO markedly increase cell surface hydrophilicity, while CA and SDS have minimal impact. HCl, NaOH, NaClO, and SDS greatly reduce the mean size of microorganisms;(ii)Compared to suspended microorganisms without treatment, CA and SDS show a limited effect on EPS and ROS production. HCl, NaOH, and NaClO increase EPS release that could accelerate biofilm formation, and trigger intracellular ROS production that could lead to the death of a large quantity of microorganisms. The concentrations of PS and PN upon exposure to HCl and NaOH increase significantly. However, the concentration of PS upon exposure to NaClO decreases;(iii)During filtration, driven by a self-protecting strategy, surviving microorganisms after exposure to HCl, NaOH, and NaClO tend to rapidly colonize onto the membrane surface, assisted by abundantly released EPS, eventually forming a living cell layer covered by dead cells, which causes a significant flux decline;(iv)The severity of membrane biofouling seems closely corelated to the intracellular ROS production and PS content in the released EPS. Under the threat of NaOH, microorganisms with the highest intracellular ROS production and PS release cause the most serious membrane fouling.

## Figures and Tables

**Figure 1 membranes-12-00920-f001:**
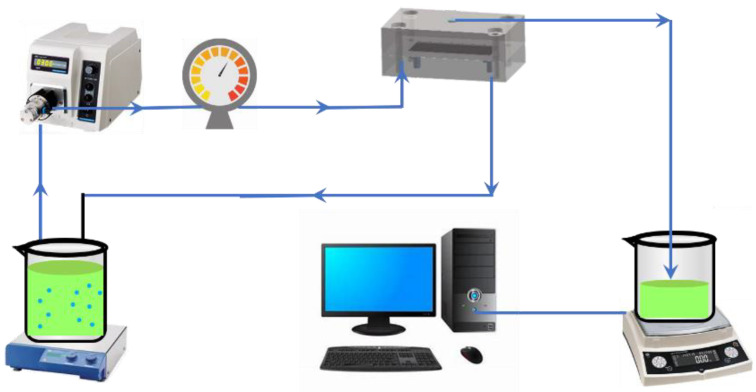
The cross-flow membrane filtration system.

**Figure 2 membranes-12-00920-f002:**
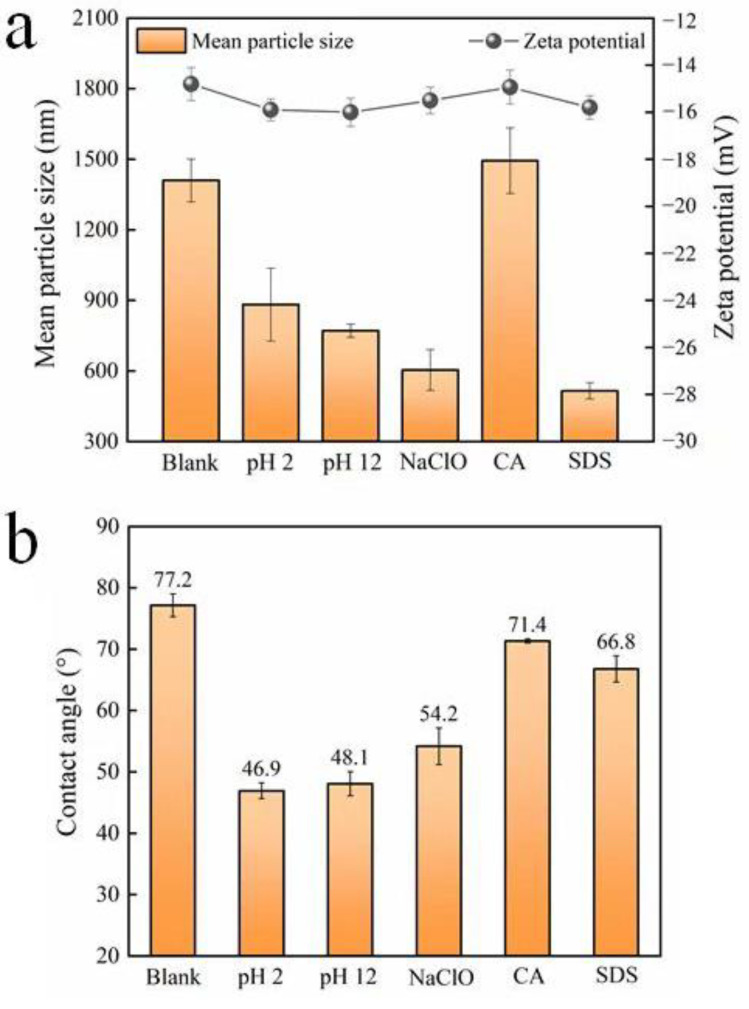
(**a**) Zeta potential and mean particle size, and (**b**) contact angle of microorganisms after exposure to different chemical cleaning agents (blank represents the sample free of chemical treatment).

**Figure 3 membranes-12-00920-f003:**
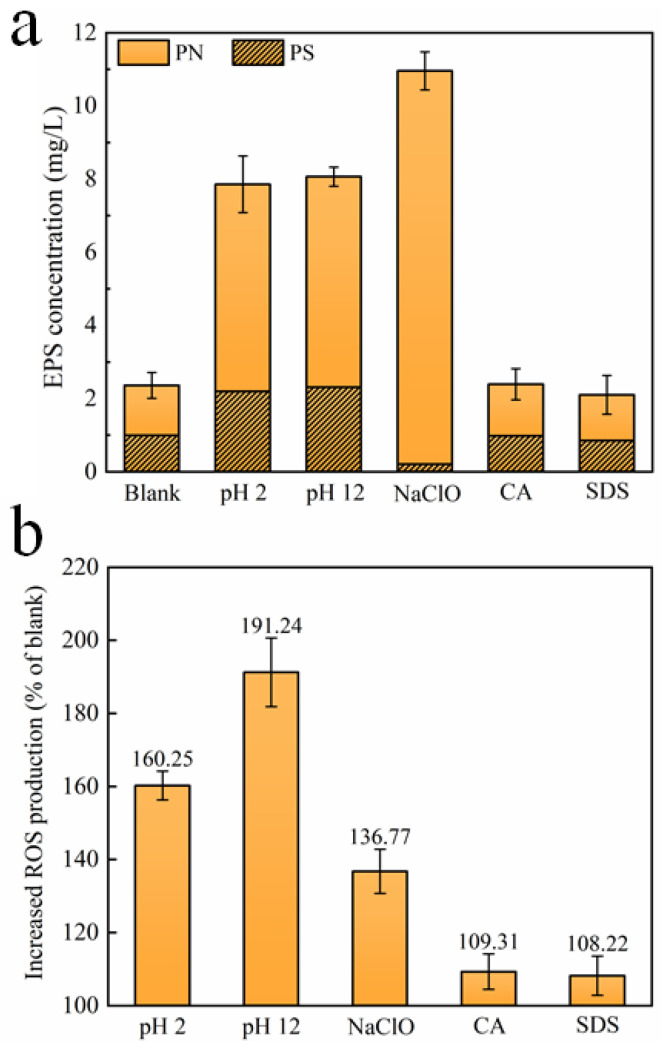
(**a**) EPS release and (**b**) ROS production of microorganisms after exposure to different chemical cleaning agents. (Blank represents the sample free of chemical treatment).

**Figure 4 membranes-12-00920-f004:**
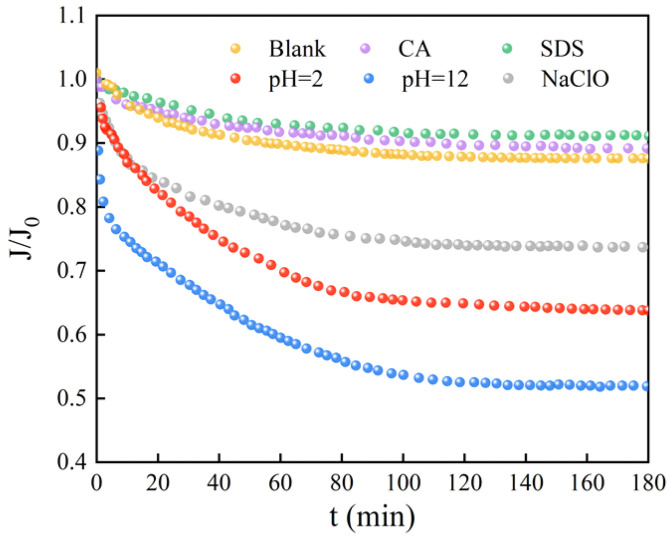
Flux decline curves of microorganisms after exposure to different chemical cleaning agents. (Blank represents the sample free of chemical treatment).

**Figure 5 membranes-12-00920-f005:**
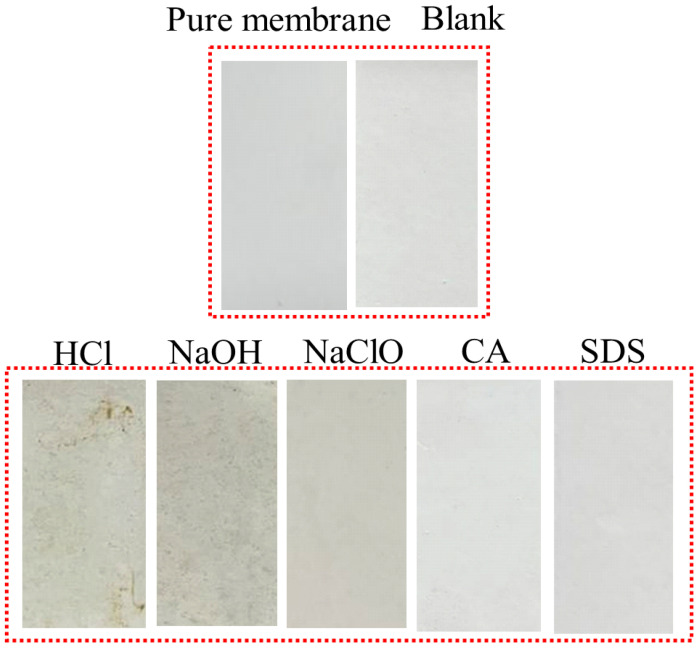
Photos of membrane surfaces after filtration of microorganisms exposed to different chemical cleaning agents. (Blank represents the sample free of chemical treatment).

**Figure 6 membranes-12-00920-f006:**
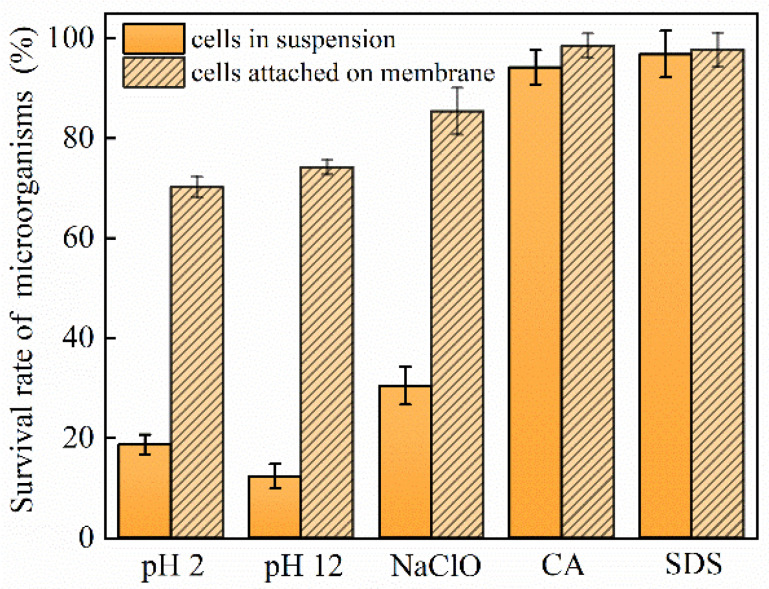
Bacterial survival rates of microorganisms after exposure to different chemical cleaning agents.

**Figure 7 membranes-12-00920-f007:**
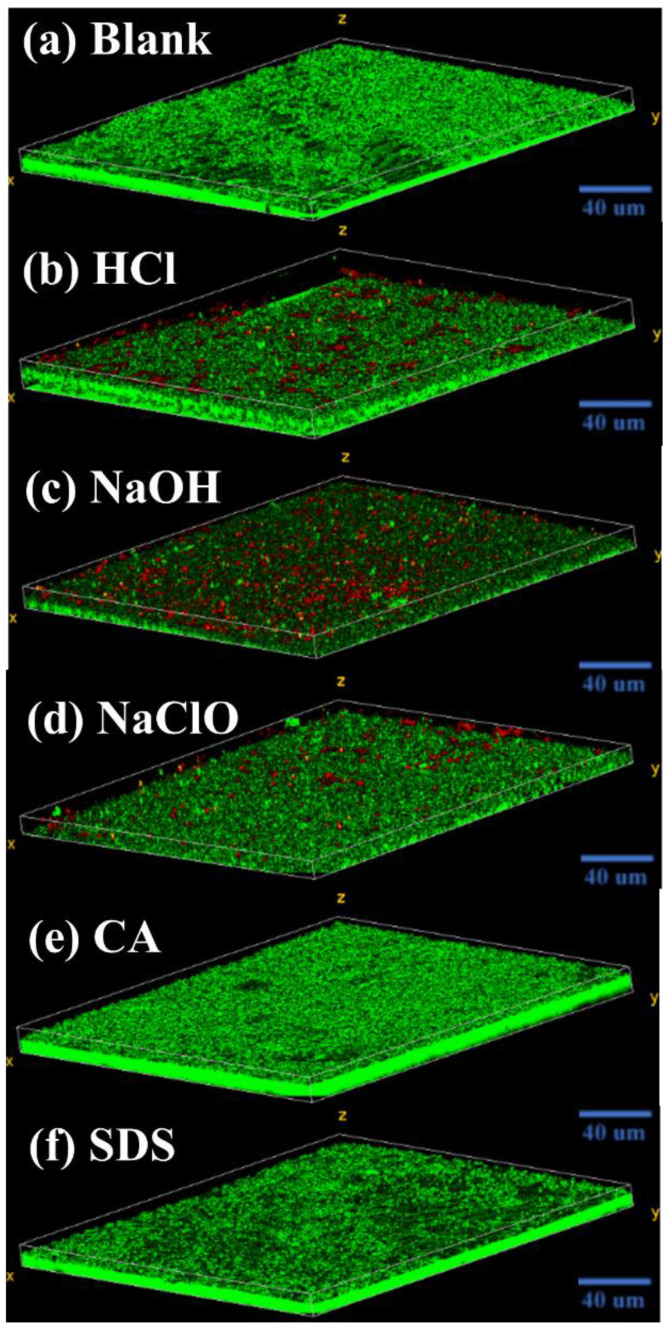
CLSM images of fouling layer on membrane surface developed from microorganisms exposed to (**a**) blank, (**b**) HCl, (**c**) NaOH, (**d**) NaClO, (**e**) CA, and (**f**) SDS. Green color: live bacteria; Red color: dead bacteria. (Blank represents the sample free of chemical treatment).

## Data Availability

Not applicable.

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
