# Peer review of "Microbial Responses to Various Types of Chemical Regents during On-Line Cleaning of UF Membranes"

_membranes, 2022, doi:10.3390/membranes12100920_

Round 1

Reviewer 1 Report

The paper is sound and address an interesting problem.

I have only detected some very minor English mistakes. Other issues on the paper realization are:

A)   Abstract and Conclusions are identic. Conclusions should be more reasoned and detailed.

B)   Figures 2 a and b are interchanged.

C)   Use the same pressure units. Do not mix MPa and Bar.

Some comments on questions I would like to see more detailed or further discussed are the following ones:

1)   Line 91. Authors say “merely a typical and commonly used concentration”. Could it be possible that different optimal concentrations were needed for the different cleaning agents?

2)   Line 102 and 103. Authors say “then resuspended in fresh PBS solution and adjusted concentration to ensure consistent biomass in the following cross-flow filtration test”. Can you be sure that resuspension could not give different foulant properties?

3)   Lines 108 & 109. It is stated thagt “(PES) membranes (Beijing Saipuruite, China) with a nominal molecular weight cut off (MWCO) of 50 kDa, average pore size of 12.18 nm and zeta potential of -3.06 mV were used in the system” Could you reference these data? At what pH was this zeta potential measured?

4)   Cross flow ultrafiltration must be better described. In particular, for example: Was crossflow made co or countercurrent? What recirculation speed was used? It seems that you use flat sheet membrane but could you describe the membrane cell?

5)   Comments made in lines 173 to 181 are confusing. Provided that charges are quite similar as confirmed by zeta potentials, the higher hydrophilicity, up to my knowledge, should reduce fouling rather than increase it. Zeta potentials correspond to negatively charged foulant as PES surface which is, except when modified, negatively charged as well. A study of the isoelectric points by measureing zeta potential at a complete pH range would allow to conclude when repulsion decreases, thus increasing, fouling.

Reviewer 2 Report

Ms. Ref. No.: membranes-1914800

Title: Microbial Responses to Various Types of Chemical Regents During On-line Cleaning of UF Membranes

Overall comment

The content of the paper is related to the journal scope and this manuscript shows novelty. The manuscript is interesting and it demonstrates thorough study. In general, the manuscript is well written and relatively easy to follow.  Current study was focused on  Microbial Responses to Various Types of Chemical Regents During On-line Cleaning of UF Membranes.

Recommendation

I found the paper to be informative and, in my view, I would recommend its acceptance after minor revision.

Main impression of article

The manuscript is interesting and it demonstrates thorough study. In general, the manuscript is well written and relatively easy to follow. Also, this manuscript shows novelty.

Journal Scope and standards

Content of this manuscript is related to journal scope and standards.

Title

Title is ok and it is appropriate.

Abstract

·  Abstract need improvement. Also, English must be improved.

·  Line 8:  chemical reagents will be employed in order.  Grammar must be improved.

·  Line 17:  reduced average particle size of suspended microorganisms: What do u mean by this sentenced?

·   

Introduction

·  Its well written

Materials and Methods

·         It is well written

Results

·   It is well written

Discussion

Its fine.

Conclusions

Fine

References

Ok

Figures

Ok

Other comments and suggestions

·   English is weak, need improvement.
